# Prevalence and risk factors of abuse against older adult women: A cross-sectional community study in Eastern Andalusia, Spain

Yolanda María de la Fuente-Robles☺, Adrián Jesús Ricoy-Cano [ID]☺*,
María Dolores Muñoz-de-Dios, Marta García-Domingo [ID]

Department of Psychology, University of Jaen, Jaen, Spain

☺ These authors contributed equally to this work and should be considered as co-first authors.
* aricoy@ujaen.es

## Abstract

Abuse against older adult women remains a serious public health issue and a flagrant violation of human rights in Spain. However, research specifically addressing abuse against older adult women is still limited, contributing to a significant gap in scientific knowledge. This study aimed to estimate the prevalence of overall abuse, and its different forms, experienced by older adult women (≥ 60 years) in a specific area of Eastern Andalusia, Spain, over the past twelve months, and to identify associated risk factors. A community-based cross-sectional study was conducted with 209 non-institutionalised older adult women, using the Geriatric Mistreatment Scale and collecting sociodemographic and lifestyle data. A high prevalence of abuse was found, with 49.3% of participants reporting some type of abuse, with psychological abuse being the most common (36.4%), followed by physical (23.0%), economic (13.9%), sexual (11.0%) and neglect (7.7%). Factors such as age, marital status, self-perceived health status and feelings of loneliness were identified as significant risk factors. These findings highlight the need to strengthen detection, prevention and response strategies within community and household settings, particularly by equipping primary care, social services and other community-based professionals—especially social workers—with the skills to identify both overt and subtle forms of abuse experienced by older adult women living at home.

## Introduction

Societies around the world are currently undergoing a global demographic transition [1]. This is clearly reflected in the structure of the population, which is seeing a significant increase in the number and proportion of people aged 60 and over [2,3]. Women constitute a majority among older adults and face particular challenges related to health, frailty and social status [4,5], which may result in increased social vulnerability

**Data availability statement:** Data availability: The complete dataset is available in the Institutional Repository of Scientific Production of the University of Jaén, accessible via the following link: https://hdl.handle.net/10953/6526.

**Funding:** This work was supported by the EXCMO. Ayuntamiento de Úbeda under Award number 2022168, Project title: Detection and analysis of situations of violence in older women in Úbeda and its surrounding areas. The author(s) also reported receiving the following financial support for authorship of this article: one author (Adrián Jesús Ricoy-Cano) was funded by the European Union through the "NextGenerationEU" program, as part of the "Grants for the Requalification of the System" program, Spanish University for 2021–2023, under the "MARGARITA SALAS" modality (Award number 26246953).

**Competing interests:** The authors have declared that no competing interests exist.

[6]. However, behind these data on population ageing lies a less visible but equally worrying reality: abuse against older adult women. Women aged 60 and over, and those with disabilities, are at increased risk of suffering abuse, yet their experiences remain largely hidden in most national and global violence-related datasets [7]. Most of the available evidence in the field has focused primarily on the issue of violence perpetrated against women of reproductive age (15–49 years), leaving a significant gap in understanding the patterns and types of abuse suffered by women aged 50 and over globally [8]. Documenting abuse against older adults—particularly older adult women—is essential, as it remains a largely overlooked public health issue, especially within domestic settings where detection is more difficult and factors such as dependency, emotional bonds, and financial constraints may heighten vulnerability [9].

Although closely related, the concepts of violence, mistreatment, and abuse are not synonymous. Despite the limitations in defining elder abuse, and the complexities this has posed for research [10,11], there is some consensus that it refers to: (1) intentional actions by a caregiver or other trusted person that cause harm or create a serious risk of harm (intentional or unintentional) to a vulnerable older person; (2) the failure on the part of a caregiver to meet the older person's basic needs or to protect them from harm [12]. Thus, elder abuse can take various forms (types of abuse): physical abuse (physical acts that cause pain or injury), psychological abuse (behaviour that causes emotional distress or psychological harm), neglect (failure of a caregiver to meet essential needs and protect the older adult from harm), economic exploitation (including deprivation or misappropriation of money and/or property) and sexual abuse (such as non-consensual sexual contact of any kind or harassment) [12–14]. These types of abuse may manifest differently in women due to a range of gender and socio-cultural factors [15], underscoring the need for a culturally specific and culturally sensitive approach to differences in identification, perception and response [10].

The concept of abuse is more restricted than that of violence against women, understood as a form of structural and gender-based violence that may result in physical, sexual, or psychological harm or suffering, including threats, coercion, or deprivation of liberty, in both public and private spaces [16,17]. In the gerontological field, the term elder mistreatment is commonly used to refer to injuries, neglect, or hazardous conditions caused—or not prevented—by another person within a relationship of responsibility or trust [12]. In this study, we adopt the more precise notion of abuse.

There is convergence in the literature regarding various factors that increase the risk of abuse against older adult women. These include marital status, age, low income, loneliness, physical and/or mental health problems, dependency in its various manifestations, difficulties in interpersonal relationships, limitations in functional capacity, depressive symptoms, dissatisfaction with quality of life and the type of relationship, among others [18–21]. Moreover, many of these factors are simultaneously recognised as vulnerability factors for victims. Their complex interplay highlights the sensitivity of older adult abuse to sociocultural and territorial variations (e.g., norms, ideologies and customs) [22], highlighting the need for contextually grounded analyses.

In Spain, the study of abuse against older adult women must be situated within a familistic sociocultural context characterised by strong intergenerational ties and a

care model largely sustained by families—and, disproportionately, by women [23]. This centrality of the household makes domestic environments particularly relevant for understanding abuse risk. Furthermore, Spain lacks a single comprehensive law explicitly addressing elder abuse; instead, protection is distributed across complementary legal frameworks. The most influential is the Organic Law 1/2004 on gender-based violence [24], while the Spanish Criminal Code [25]—particularly Article 173 and related provisions—penalises degrading treatment and physical or psychological harm. Yet, the absence of a legal definition of "elder abuse" contributes to fragmented protections.

In relation to the epidemiology of abuse, previous international meta-analysis reviews have detected varying incidences of abuse against women aged 60 and over; for example, Yon et al. [26] detected an estimated global prevalence of 14.1%, Zhang et al. [27], 36.0%; and, more recently, Ricoy-Cano et al. [28], 27.3%, the latter two specifically in rural and/or remote settings.

More specifically, in the case of Spain, although considered an advanced country in enacting laws and policies to address gender-based violence [29], the evidence available on accurate and representative data on older adult abuse has been rather limited to date. Nevertheless, some previous studies had already warned of the problem. A study in Southeastern Spain reported a 44.6% suspicion rate of abuse in older adults [30]. Meanwhile, research carried out in Girona (Northeastern Spain) found a prevalence of abuse of 29.3% [31]. Another European multilevel study [32] estimated that, in Spain, 15.9% of older adult women had experienced some form of abuse, including 12.8% psychological abuse, 5.5% financial abuse, and 1.6% physical or sexual abuse with injury, highlighting the persistence of the problem despite its comparatively lower magnitude relative to other European countries. Likewise, a comparative investigation conducted in Córdoba (Spain), Santa Cruz de la Sierra (Bolivia), and the Azores (Portugal) identified suspected abuse in 6.9% of Spanish older adults, compared with 39% in Bolivia and 24.5% in Portugal [33]. Similarly, the comparative study by Farnia et al. [34] between Spain and Iran showed that 39.1% of older adults assessed in Spain presented indications of abuse—figures considerably lower than those observed in Iran (80.5%). Finally, aside from several broader European reviews [35,36] that have referred to the Spanish case, the available literature remains limited and offers little disaggregated data, leaving substantial gaps in understanding the magnitude and characteristics of abuse against older adult women in the country.

More evidence is needed to understand the extent and variability of older adult abuse in Spain, with particular attention to its manifestations against older adult women and to the importance of culturally and territorially situated analyses.

Given that abuse of older adults is a multifactorial phenomenon, no single approach can fully explain its causes. The review of etiological theories by Abolfathi et al. [37] shows that risk arises from the interplay of individual, relational, community, and sociocultural factors, including mutual dependency (social exchange), caregiver stress (situational and role accumulation theories), patriarchal dynamics (feminist theory), structural marginalization (political economy), caregiver psychopathology, and cultural meanings associated with ageing (symbolic interactionism). Guided by this evidence, this study adopts a broad ecological perspective to situate abuse against older adult women within a web of interconnected vulnerabilities in home and community settings, particularly in families with strong expectations of care.

Taken together, these elements underscore the importance of examining abuse specifically against older adult women in Spain, where sociocultural norms, care structures, and gender inequalities intersect to shape exposure and vulnerability. Therefore, this study aimed to assess the prevalence of abuse against older adult women, as well as its different forms—including physical, psychological, economic and sexual abuse, and neglect—perpetrated by any person within their environment over the past twelve months. In addition, the study explored socio-demographic and lifestyle factors associated with these experiences of abuse, identifying possible risk factors.

## Materials and methods

### Study design, sites and participants

This cross-sectional community-based study was conducted from 15 June to 31 July 2023 in an urban area and several rural sites in Eastern Andalusia, Spain, including *pedanías* (infra-municipal entities). Non-institutionalised women ≥ 60

years old, living in their own homes or with family members, who did not suffer from dementia or other severe cognitive disabilities, or communication problems that made it impossible to transmit information on their own, were included. Municipalities and various women's associations facilitated contact with potential participants. A few weeks beforehand, the collaborating institutions and entities received a brief letter presenting the research team and the study, so that this information could be shared with potential participants in these safe spaces. This was done as a protective measure to avoid inducing or aggravating any exposure to abuse.

In this context, the study's participants were selected using a non-probability, purposive sampling method, based on the networks and dissemination capacity of the collaborating institutions. With a high response rate (76.8%), the final sample consisted of 209 participants. The sample size was defined by feasibility criteria, considering the study's time frame and available recruitment channels. After providing informed consent, all participants were interviewed in private by members of the research team and trained assistants with expertise in gender and elder abuse, using a face-to-face survey technique to ensure clarity and response accuracy. Most interviews were conducted in women's associations, or neighbourhood organisations, in private rooms designated for this purpose. For participants with reduced mobility, interviews were conducted at home. Municipal social services, specifically through the home help service provider, supported the process by facilitating initial contact with potential participants. These interviews were scheduled to ensure privacy, ideally at times when participants were alone, in order to maintain confidentiality and avoid interruptions. No proxy interviews were required. Data collection was carried out by the research team and assistants using a structured questionnaire. When participants experienced physical difficulties in completing the form independently, responses were entered in real time into a digital version of the questionnaire (Google Forms) by the interviewer. In other cases, answers were recorded on paper and later digitised by support staff for integration into the online system and subsequent data management. The interviews lasted approximately 40 minutes.

## Measures

### Geriatric Mistreatment Scale

Abuse among participants was assessed using the Spanish version of the Geriatric Mistreatment Scale [38], a 22-item instrument that assesses physical, psychological, neglect, economic and sexual abuse in the preceding 12 months. Prior to the questions a brief introduction was presented to the participants, which read "Difficult situations are known to exist that are often not discussed but significantly affect older adults. Understanding what happens will allow taking necessary measures to prevent them from occurring in the future. Please let me know if you have experienced any of the following problems in the past year, either at home or outside." During the last 12 months you... An example item is: "Have you been humiliated or made fun of?" Participants responded using a dichotomous scale, which was coded as "0 = No" (indicating no occurrence of abuse in the last 12 months), and "1 = Yes" (indicating occurrence of abuse in the last 12 months). The instrument enables the computation of both an overall measure of abuse and specific subtypes. Participants were classified as having experienced overall abuse if they responded affirmatively to at least one of the 22 items. Subtypes of abuse were assessed by grouping relevant items as follows:

- Physical abuse: Includes actions intended to cause physical pain or injury (e.g., being hit or pushed).

- Psychological abuse: Involves verbal aggression, humiliation, intimidation, or threats that cause emotional distress.

- Economic abuse: Encompasses situations where money or belongings are taken without consent, or the individual is subjected to financial coercion.

- Sexual abuse: Covers any unwanted sexual contact or harassment.

- Neglect: Refers to the failure to provide necessary care, including assistance with daily activities, medical attention, hygiene, or nutrition.

Each subtype was measured by summing affirmative responses to the corresponding items. A participant was considered to have experienced a specific form of abuse if they responded "yes" to at least one item within that subtype. In its validation study [38] this instrument showed good psychometric properties and internal consistency ($\alpha = .83$). The version of this scale has already been used in previous studies with samples of older adults [39,40]. For the present study, the research team carried out a brief content review to ensure suitability for the Andalusian context, and a pilot test with 11 older adult women confirmed that the items were well understood and culturally appropriate, requiring no substantial modifications. Internal consistency of the scale was assessed using McDonald's Omega, calculated in RStudio ($\omega = .69$; $\omega_h = .55$), suggesting moderate internal consistency. This level of reliability is consistent with the multidimensional structure of the instrument, which assesses five conceptually distinct but related subtypes of abuse (physical, psychological, economic, sexual, and neglect). In such heterogeneous constructs, moderate omega values are expected and reflect the absence of a strong unidimensional latent factor rather than measurement inadequacy. Because only binary subtype-level indicators (rather than item-level responses) were available, subtype-specific reliability estimates could not be computed. Accordingly, the use of a combined "any abuse" outcome variable was supported by both conceptual and empirical considerations. First, phi ($\varphi$) correlations among abuse subtypes ranged from 0.06 to 0.47, with the strongest associations observed among psychological, economic, sexual, and neglect dimensions (e.g., $\varphi = 0.47$ for neglect–sexual; $\varphi = 0.37$ for psychological–sexual; $\varphi = 0.33$ for psychological–financial), supporting the coherence of a composite abuse indicator. Second, among the 103 women who reported any form of abuse (49.3%), 55.3% experienced two or more forms simultaneously, indicating a substantial pattern of polyvictimisation that further supports the analytical use of a combined outcome.

## Socio-demographic information

The research team designed a survey form including socio-demographic variables that could potentially influence the risk of abuse in older adults —particularly in women, based on previous research [31,41–43], and with sensitivity to the intrinsic social and cultural attributes of the areas and communities under study. These variables were selected and pilot-tested to ensure their appropriateness and validity in the specific context of this study. Several variables were subsequently recoded to improve statistical power and interpretability. Age was initially recorded in five-year intervals (from 60 to 95+ years) and was recoded into three categories (coded as "0 = 60–69," "1 = 70–79," "2 = 80 or older"). Place of residence was self-reported and categorised as "0 = urban; 1 = rural," based on the population size of the municipality. In Spain, rural areas are defined as villages or hamlets or neighbourhoods located in municipalities with fewer than 10,000 inhabitants. Educational level was originally recorded as low, middle or high. For analytical purposes, it was recoded as "0 = low; 1 = middle-high." The "low" category grouped participants with basic literacy or primary education, acknowledging some heterogeneity in formal attainment. The "middle-high" category included secondary, post-secondary non-tertiary, and higher education levels. Marital status, initially captured in six categories, was recoded into "0 = married" versus "1 = widowed/divorced/single/separated." Income level, originally recorded in four ordinal categories, was dichotomised as "0 = not enough; 1 = enough," using the national Multieffect Income Indicator (IPREM) of the corresponding year as a reference threshold. Housing was recoded as "0 = own; 1 = rental." Living arrangements were simplified to "0 = alone; 1 = other family members," combining various family structures into a single category. Self-perceived general health, originally assessed using five levels, was recoded as "0 = dissatisfied; 1 = neither satisfied nor dissatisfied; 2 = satisfied." Other variables such as dependency situation by age, illness or disability, participation in community activities, social services use in the past 12 months, and self-perceived feeling of loneliness retained their original dichotomous format ("0 = no; 1 = yes"). The recoding strategy aimed to enhance the clarity and relevance of statistical analysis while preserving conceptual integrity. These recoding decisions were guided by three considerations: (a) ensuring sufficient cell frequencies for valid chi-square and logistic regression analyses given the total sample size of $n = 209$; (b) aligning with the categorisations employed in

comparable studies of older adult abuse in Spain and internationally, which facilitates cross-study comparability; and (c) reflecting contextually and policy-relevant thresholds. Specifically, the binary age split at 80 years is consistent with the WHO classification of "old-old" adults and is widely used in gerontological and elder abuse research. The income dichotomisation is referenced to the IPREM, the standard Spanish social policy threshold for defining economic insufficiency, ensuring that the resulting categories carry direct contextual meaning. While broader categories entail some loss of statistical precision relative to continuous or polytomous measures, this trade-off was considered necessary and appropriate given the sample size and the exploratory nature of the study.

## Statistical analyses

Data were analysed using SPSS® Statistics for MacOS, version 29.0.2.0 [44]. Survey responses were exported from the digital form as a CSV file and imported into SPSS. For participants who completed the questionnaire on paper, trained research assistants manually entered the responses into the digital database. To minimise data-entry bias, a double-entry verification protocol was applied, and random checks were conducted on 10% of the entries. The analysis was carried out in three main steps. First, a univariate analysis was performed to present the sociodemographic and lifestyle characteristics of the participants. Second, cross-tabulations were used to perform a bivariate analysis between socio-demographic and lifestyle characteristics and the different forms of abuse, and $Chi^2$ statistics were calculated in the search for possible associations. In performing the $Chi^2$ test, we followed the guideline whereby the expected frequencies for each category should be at least 1, and furthermore, no more than 20.0% of the categories should have expected frequencies lower than 5 [45,46]. In instances where these assumptions were not met, Fisher's Exact Test was applied instead.

Finally, an adjusted multivariable binary logistic regression analysis was performed to examine different risk factors for abuse. The dependent variables were abuse in any form (physical, psychological, economic, sexual or neglect) over the last 12 months, while the independent variables were sociodemographic and lifestyle characteristics. A backward stepwise (likelihood ratio) logistic regression approach was employed to identify the most parsimonious model. Variables were sequentially removed based on lack of statistical significance ($p > 0.05$), allowing retention of those with the strongest associations while minimising overfitting. A 95% confidence interval (CI) and a p-value cut-off of 0.05 were applied in all regression models. This approach was selected in an exploratory framework, given the absence of a fully established theoretical model specifying which sociodemographic and lifestyle risk factors operate in the specific sociocultural context of older adult women in Eastern Andalusia. The backward elimination method allows the empirical data to guide variable selection while maintaining a parsimonious final model, and is consistent with comparable prevalence and risk factor studies in the elder abuse literature. The methodological limitations of this strategy are discussed in the Limitations section.

Prior to model estimation, multicollinearity was assessed using tolerance and variance inflation factor (VIF), with all values falling within acceptable thresholds (VIF range: 1.03–1.40; Tolerance range: 0.71–0.97). Influential cases were examined using Cook's Distance, with no values exceeding the recommended threshold ($D > 1$), hence all cases were retained [47]. Additionally, the model's discriminative capacity was evaluated by calculating the area under the receiver operating characteristic curve (AUROC), with concordance statistics (c-statistics) reported for each dependent variable. AUROC values ranged from 0.67 (neglect) to 0.82 (any form of abuse), indicating fair to good discriminative power of the logistic regression models. This approach is recommended to assess the predictive performance of logistic regression models [48]. No missing data were observed for the variables included in the regression models; therefore, the full sample of 209 participants was retained in all analyses.

## Ethical considerations

The protocol for this study was approved by the Ethics Committee on Human Research (CEIH) of the University of Jaén (Protocol Code:MAY.23/1PRY). All participants were duly informed of the purpose of the research, the voluntary nature of

the process, the confidentiality of their participation, their rights, and the handling of the information provided, prior to the start of the study. They were provided with an informed consent form, and consent was obtained verbally and/or in writing. Verbal consent was recorded in an online form, whereas written consent was documented through the signed information sheet. Participants were informed that they had the right to refuse to answer any question, to terminate the interview at any time without penalty, and that they could also withdraw their participation even after the interview had concluded. The recruitment period for this study began on June 15 and ended on July 31, 2023.

## Results

### Socio-demographic and lifestyle characteristics

Table 1 describes the socio-demographic and lifestyle characteristics of the sample of older adult women surveyed. Two hundred and nine women aged 60 and over participated in this study. The most representative group in terms of age range was 80 years and older, with 35.9% of the total number of participants. In terms of place of residence, the majority lived in urban areas, constituting 86.1% of the sample. In terms of educational level, 82.8% had a low level of education. 60.3% of the participants were widowed, divorced, single or separated. 80.9% of these women indicated that they did not have sufficient income, although the vast majority (88.5%) were homeowners. In relation to their cohabitation status, 53.1% lived with other members of the family. 63.6% of the women had no dependency. Only 41.4% were involved in community activities. Only 24.4% of the participants indicated that they had used social services in the last 12 months. Almost half of these women (44.0%) reported feeling lonely, and 38.3% reported feeling dissatisfied with their health.

### Overall prevalence of abuse and by type of abuse

As shown in Table 2, more than 49% of the participating older adult women reported having experienced abuse (in any form) at least once over the last 12 months. Regarding the specific types of abuse in the last 12 months, the following prevalences were found: 23.0%, physical; 36.4%, psychological; 13.9%, economic; 11.0%, sexual and 7.7%, neglect.

### Abuse and socio-demographic/lifestyle variables

As can be seen in Table 2, abuse in any form was most prevalent for the 60–69 (53.0%) and 70–79 (58.8%) age groups, and least prevalent for women aged 80 and over (37.3%). Compared to married women, abuse in any form was more common among widowed/divorced/single/separated women (41.0% vs. 54.8%, respectively). Analyses also showed that abuse in any form was more common among women who were not dependent (54.9%) compared to those who were dependent (39.5%); and for those who perceived themselves to be in neither satisfied nor dissatisfied health (62.1%) rather than dissatisfied (43.8%) or satisfied (42.9%).

Women with a middle to high levels of education compared to those with low levels were more likely to suffer physical abuse (36.1% vs. 20.2%). 31.5% of older adult women who reported feeling lonely reported physical abuse, compared to 16.2% of those who did not feel lonely.

As for psychological abuse, a higher prevalence was observed in women aged 70–79 (44.1%), followed by those aged 60–69 (40.9%), while the proportion was lower in women ages 80 and older. Among women who felt lonely, psychological abuse was more frequent (44.6%) than among those who did not (29.9%), in a way similar to those who perceived their health status as neither satisfied nor dissatisfied (48.5%), compared to those who were dissatisfied or satisfied with their health status.

Economic abuse was more common among those participants who used social services in the last 12 months (23.5%) compared to those who did not (10.8%). This type of abuse was also higher among those who felt lonely (19.6%) compared to those who did not (9.4%).

**Table 1. Sociodemographic and lifestyle characteristics of the participants.**

| Variables | Number (%) |
| --- | --- |
| *Age range* | |
| 60-69 | 66 (31.6%) |
| 70-79 | 68 (32.5%) |
| 80 or older | 75 (35.9%) |
| *Place of residence* | |
| Urban | 180 (86.1%) |
| Rural | 29 (13.9%) |
| *Education level* | |
| Low | 173 (82.8%) |
| Middle-high | 36 (17.2%) |
| *Marital status* | |
| Married | 83 (39.7%) |
| Widowed/divorced/single/separated | 126 (60.3%) |
| *Income level* | |
| Not enough | 169 (80.9%) |
| Enough | 40 (19.1%) |
| *Housing* | |
| Own | 185 (88.5%) |
| Rental | 24 (11.5%) |
| *Living arrangements* | |
| Alone | 98 (46.9%) |
| Other family members | 111 (53.1%) |
| *Dependency situation* [a] | |
| Yes | 76 (36.4%) |
| No | 133 (63.6%) |
| *Participation in community activities* | |
| Yes | 86 (41.4%) |
| No | 123 (58.9%) |
| *Social Services use (last 12 months)* | |
| Yes | 51 (24.4%) |
| No | 158 (75.6%) |
| *Feeling of loneliness (self-perceived)* | |
| Yes | 92 (44.0%) |
| No | 117 (56.0%) |
| *General health (self-perceived)* | |
| Dissatisfied | 80 (38.3%) |
| Neither satisfied nor dissatisfied | 66 (31.6%) |
| Satisfied | 63 (30.1%) |

[a]Dependency situation (by age, illness or disability).

Participants who were not dependent experienced sexual abuse more frequently than those who were (14.3% vs. 5.3%). Sexual abuse was also more common among those participants who reported feeling lonely.

Dependent older adult women reported lower levels of neglect (1.3%) than those who were not (11.3%).

**Table 2. Sociodemographic and lifestyle variables associated with the prevalence of abuse (physical, psychological, economic, sexual and neglect) in the last 12 months.**

| Variables | Abuse (AF) n (%) | Physical n (%) | Psychological n (%) | Economic n (%) | Sexual n (%) | Neglect n (%) |
|---|---|---|---|---|---|---|
| Total | 103 (49.3) | 48 (23.0) | 76 (36.4) | 29 (13.9) | 23 (11.0) | 16 (7.7) |
| *Age range* | | | | | | |
| 60-69 (31.6%) | 35 (53.0) | 16 (24.2) | 27 (40.9) | 10 (15.2) | 9 (13.6) | 6 (9.1) |
| 70-79 (32.5%) | 40 (58.8) | 17 (25.0) | 30 (44.1) | 9 (13.2) | 9 (13.2) | 8 (11.8) |
| 80 or older (35.9%) | 28 (37.3) | 15 (20.0) | 19 (25.3) | 10 (13.3) | 5 (6.7) | 2 (2.7) |
| *p* value | 0.028* | 0.744 | 0.049* | 0.936 | 0.324 | 0.108 |
| *Place of residence* | | | | | | |
| Urban (86.1%) | 90 (50.0) | 41 (22.8) | 66 (36.7) | 29 (16.1) | 20 (11.1) | 13 (7.2) |
| Rural (13.9%) | 13 (44.8) | 7 (24.1) | 10 (34.5) | 0 | 3 (10.3) | 3 (10.3) |
| *p* value | 0.605 | 0.872 | 0.821 | NS | 0.903 | 0.557 |
| *Education level* | | | | | | |
| Low (82.8%) | 83 (48.0) | 35 (20.2) | 60 (34.7) | 21 (12.1) | 19 (11.0) | 13 (7.5) |
| Middle-high (17.2%) | 20 (55.6) | 13 (36.1) | 16 (44.4) | 8 (22.2) | 4 (11.1) | 3 (8.3) |
| *p* value | 0.408 | 0.039* | 0.268 | 0.111 | 0.982 | 0.866 |
| *Marital status* | | | | | | |
| Married (39.7%) | 34 (41.0) | 19 (22.9) | 25 (30.1) | 11 (13.3) | 7 (8.4) | 4 (8.4) |
| Widowed/divorced/single/separated (60.3%) | 69 (54.8) | 29 (23.0) | 51 (40.5) | 18 (14.3) | 16 (12.7) | 12 (9.5) |
| *p* value | 0.050* | 0.983 | 0.128 | 0.833 | 0.335 | 0.211 |
| *Income level* | | | | | | |
| Not enough (80.9%) | 83 (49.1) | 37 (21.9) | 61 (36.1) | 23 (13.6) | 19 (11.2) | 13 (7.7) |
| Enough (19.1%) | 20 (50.0) | 11 (27.5) | 15 (37.5) | 6 (15.0) | 4 (10.0) | 3 (7.5) |
| *p* value | 0.920 | 0.448 | 0.868 | 0.819 | 0.821 | 0.967 |
| *Housing* | | | | | | |
| Own (88.5%) | 89 (48.1) | 42 (22.7) | 68 (36.8) | 24 (13.0) | 18 (9.7) | 14 (7.6) |
| Rental (11.5%) | 14 (58.3) | 6 (25.0) | 8 (33.3) | 5 (20.8) | 5 (20.8) | 2 (8.3) |
| *p* value | 0.346 | 0.801 | 0.743 | 0.295 | 0.102 | 0.894 |
| *Living arrangements* | | | | | | |
| Alone (46.9%) | 54 (55.1) | 23 (23.5) | 40 (40.8) | 15 (15.3) | 14 (14.3) | 10 (10.2) |
| Other family members (53.1%) | 49 (44.1) | 25 (22.5) | 36 (32.4) | 14 (12.6) | 9 (8.1) | 6 (5.4) |
| *p* value | 0.114 | 0.871 | 0.209 | 0.574 | 0.154 | 0.193 |
| *Dependency situation* | | | | | | |
| Yes (36.4%) | 30 (39.5) | 13 (17.1) | 23 (30.3) | 8 (10.5) | 4 (5.3) | 1 (1.3) |
| No (63.6%) | 73 (54.9) | 35 (26.3) | 53 (39.8) | 21 (15.8) | 19 (14.3) | 15 (11.3) |
| *p* value | 0.032* | 0.128 | 0.166 | 0.290 | 0.045* | 0.009** |
| *Participation in community activities* | | | | | | |
| Yes (41.4%) | 47 (54.7) | 19 (22.1) | 35 (40.7) | 13 (15.1) | 10 (11.6) | 9 (10.5) |
| No (58.9%) | 56 (45.5) | 29 (23.6) | 41 (33.3) | 16 (13.0) | 13 (10.6) | 7 (5.7) |
| *p* value | 0.194 | 0.802 | 0.276 | 0.664 | 0.810 | 0.201 |
| *Social Services use (last 12 months)* | | | | | | |
| Yes (24.4%) | 25 (49.0) | 10 (19.6) | 20 (39.2) | 12 (23.5) | 9 (17.6) | 4 (7.8) |
| No (75.6%) | 78 (49.4) | 38 (24.1) | 56 (35.4) | 17 (10.8) | 14 (8.9) | 12 (7.6) |
| *p* value | 0.966 | 0.512 | 0.626 | 0.022* | 0.081 | 0.954 |
| *Feeling of loneliness (self-perceived)* | | | | | | |
| Yes (44.0%) | 50 (54.3) | 29 (31.5) | 41 (44.6) | 18 (19.6) | 16 (17.4) | 9 (9.8) |

*(Continued)*

**Table 2.** (Continued)

| Variables | Abuse (AF) n (%) | Physical n (%) | Psychological n (%) | Economic n (%) | Sexual n (%) | Neglect n (%) |
|---|---|---|---|---|---|---|
| No (56.0%) | 53 (45.3) | 19 (16.2) | 35 (29.9) | 11 (9.4) | 7 (6.0) | 7 (6.0) |
| p value | 0.194 | 0.009** | 0.029* | 0.035* | 0.009** | 0.305 |
| *General health (self-perceived)* | | | | | | |
| Dissatisfied (38.3%) | 35 (43.8) | 15 (18.8) | 25 (31.3) | 12 (15.0) | 7 (8.8) | 3 (3.8) |
| Neither satisfied nor dissatisfied (31.6%) | 41 (62.1) | 15 (22.7) | 32 (48.5) | 7 (10.6) | 10 (15.2) | 7 (10.6) |
| Satisfied (30.1%) | 27 (42.9) | 18 (28.6) | 19 (30.2) | 10 (15.9) | 6 (9.5) | 6 (9.5) |
| p value | 0.041* | 0.382 | 0.046* | 0.642 | 0.424 | 0.241 |

AF = any form; NS = non-significant.

** $p$ = 0.01; * $p$ = 0.05

In addition to the significant findings, several variables presented *p*-values close to the conventional 0.05 threshold, suggesting potential associations that, although not statistically significant, may hold substantive relevance. Small elevations in prevalence were observed for abuse (any form) according to living arrangements, participation in community activities and feeling of loneliness (self-perceived). For neglect according to age ($p$ = 0.108), sexual abuse in relation to housing status ($p$ = 0.102), and economic abuse in relation to education level ($p$ = 0.111). Living arrangements also showed near-threshold differences for overall abuse ($p$ = 0.114) and sexual abuse ($p$ = 0.154). For physical and psychological abuse, dependency status showed suggestive patterns ($p$ = 0.128 and $p$ = 0.166, respectively), while the use of social services approached significance for sexual abuse ($p$ = 0.081). Finally, both community participation and self-perceived loneliness displayed borderline associations with overall abuse ($p$ = 0.194).

### Risk factors for abuse against older adult women

A logistic regression analysis (backward method) was used to identify the risk factors associated with the abuse against older adult women, considering the combined effect of the variables. All socio-demographic and lifestyle variables were entered into the initial model for each type of abuse, including abuse in any form.

This analysis showed that being age 60–69 (OR = 2.356; 95% CI: 1.113–4.991) or 70–79 (OR = 2.866; 95% CI: 1.400–5.865), widowed/divorced/single or separated (OR = 2.249; 95% CI: 1.212–4.174), or reporting a self-perceived general health status of "neither satisfied nor dissatisfied" (OR = 2.359; 95% CI: 1.133–4.910) were risk factors for abuse in any form. After adjusting for all other variables in the model, women aged 60–69 had more than twice the odds of experiencing overall abuse compared to those aged 80 or older, and women aged 70–79 showed nearly 187% higher odds. Similarly, women in widowed/divorced/single/separated categories had more than double the odds of reporting any form of abuse compared to married women.

For physical abuse, feeling lonely (OR = 2.327; 95% CI: 1.196–4.527) was identified as a risk factor, indicating that, after controlling for the remaining predictors, women who reported loneliness had more than double the odds of experiencing physical abuse compared to those who did not report loneliness.

Being in the 60–69 age group (OR = 2.469; 95% CI: 1.149–5.307) or 70–79 (OR = 2.739; 95% CI: 1.316–5.700), or being widowed/divorced/single or separated (OR = 2.006; 95% CI: 1.064–3.781), and experiencing feelings of loneliness (OR = 1.951; 95% CI: 1.079–3.528), were found to be risk factors for psychological abuse. Specifically, after adjusting for all variables, women aged 60–69 or 70–79 had more than twice the odds of reporting psychological abuse than those aged 80 or older.

Finally, using social services in the last month (OR = 2.757; 95% CI: 1.165–6.525) and experiencing loneliness (OR = 2.867; 95% CI: 1.099–7.482) were risk factors for economic and sexual abuse, respectively. Not being in a dependent situation (OR = 3.362; 95% CI: 1.059–10.674) was also identified as a sexual abuse risk factor.

As shown in Table 3, this regression analysis confirmed most of the significant associations previously detected in the univariate analysis, reinforcing the validity of the findings and underlining the robustness of the relationships identified.

## Discussion

This study examined abuse prevalence and its associated risk factors among older adult women in Eastern Andalusia, alongside the socio-demographic and lifestyle variables linked to these experiences. Despite the existence of previous studies carried out in Spain, the data provided in this research support and expand on previous findings on the preva-lence of abuse and its associated factors, also at the international level. In this study we found that almost half of the older adult women participating had suffered at least one episode of abuse in some form (49.3%) in the last year; more specifically, 23% of the participants reported physical abuse; 36.4%, psychological; 13.9%, economic; 11%, sexual abuse; and 7.7%, neglect. This prevalence figure should be interpreted in light of three methodological considerations. First, the GMS functions as a screening instrument designed to detect the possible presence of abuse rather than to provide clinically confirmed case rates; it captures a broad range of experiences, including those that may be sub-threshold or ambiguous. Second, the classification criterion used in the present study—namely, any affirmative response to at least one of the 22 items—is intentionally inclusive and likely identifies experiences that narrower, case-confirmation protocols would not record. Third, the non-probability purposive sampling through women's associations and social services may have reached women with heightened awareness of abusive experiences or those already engaged in support networks, which could contribute to higher self-reporting rates. Taken together, these factors likely explain the apparently high overall prevalence and should be considered when comparing our estimates with those from studies using different measurement approaches, sampling frames, or abuse definitions.

**Table 3. Risk factors for the abuse against older adult women over the last 12 months (logistic regression analysis-backward method).**

| Type of abuse | Risk factor | OR | %95 CI | | p-Value |
| --- | --- | --- | --- | --- | --- |
| | | | Lower | Upper | |
| Abuse (AF) | Age range (60–69) | 2.356 | 1.113 | 4.991 | 0.025* |
| | Age range (70–79) | 2.866 | 1.400 | 5.865 | 0.004** |
| | Marital status (W/D/S/S) | 2.249 | 1.212 | 4.174 | 0.010** |
| | General health (neither satisfiednor dissatisfied) | 2.359 | 1.133 | 4.910 | 0.022* |
| Physical | Feeling of loneliness (yes) | 2.327 | 1.196 | 4.527 | 0.013* |
| Psychological | Age range (60–69) | 2.469 | 1.149 | 5.307 | 0.021* |
| | Age range (70–79) | 2.739 | 1.316 | 5.700 | 0.007** |
| | Marital status (W/D/S/S) | 2.006 | 1.064 | 3.781 | 0.031* |
| | Feeling of loneliness (yes) | 1.951 | 1.079 | 3.528 | 0.027* |
| Economic | Social services use in the lastmonth (yes) | 2.757 | 1.165 | 6.525 | 0.021* |
| Sexual | Dependency situation (no) | 3.362 | 1.059 | 10.674 | 0.040* |
| | Feeling of loneliness (yes) | 2.867 | 1.099 | 7.482 | 0.031* |

*Note.* AF = any form; W/D/S/S = widowed/divorced/single/separated; OR = odds ratio; CI = confidence interval. Reference variables: Age range (80 or older); Marital status (married); Dependency situation (yes); Feeling of loneliness (no); Use of social services in the last month (no); General health (Satisfied).

** *p* = 0.01; * *p* = 0.05

Regarding associated factors, significant associations were observed for age and dependency status with any-form abuse; educational level and loneliness with physical abuse; loneliness with psychological abuse; social service use and loneliness with economic abuse; dependency status and loneliness with sexual abuse; and loneliness with neglect. Although not statistically significant, additional trends suggested potential links between abuse and living arrangements, participation in community activities, and loneliness. These patterns are consistent with previous studies indicating that limited social participation and higher levels of perceived loneliness increase vulnerability to abuse among older adults [49,50], while differing from research consistently reporting higher risk among highly dependent older adults [51,52], suggesting that dependency in this sample may interact with other psychosocial factors rather than operate as a uniform predictor.

Taken together, these patterns align with the ecological and critical-ecological perspectives [37,53], insofar as the interplay between individual vulnerabilities (e.g., loneliness, dependency), relational conditions, and broader sociocultural factors reflects a multifactorial configuration of risk rather than isolated determinants.

Previous community-based cross-sectional studies in European countries, including Spain, found similar or even lower prevalences for various types of abuse against older adults compared to the results obtained in our study. For example, Eslami et al. [43] found a 34% psychological abuse rate and an 11.5% physical abuse rate in a sample of older adult European women and men. Similarly, Lindert et al. [54] reported a prevalence of 18.9% for psychological abuse, 1% for sexual abuse, 3.7% for economic abuse, and 2.3% for neglect, specifically among women. However, higher frequencies of abuse against older adult women can also be found in countries of the Global South, in areas such as Iran (90.4%) [55], Kenya (82.1%) [56] and Brazil (54.3%) [57].

In a geographical and socio-cultural environment proximate to that of our study, southeastern Spain, a study conducted more than a decade ago revealed a worrying rate of suspected elder abuse, reaching 44.6% [30]. This figure, although slightly lower, is remarkably similar to our study's findings. In a study carried out in Girona (northeast Spain), however, a significantly lower prevalence of any type of suspected abuse was recorded: 29.3% [31]. These discrepancies in prevalence rates according to geographical location underline the persistence of the problem and its sensitivity to territorial factors [58,59], in addition to these women's possible difficulties as identifying themselves as victims of abuse [29]. Thus, as previous studies have suggested, the field of research on abuse against older adult women may be overlooking a larger population of survivors than previously thought [60].

In line with our findings, the literature in the field often detects psychological/emotional abuse as one of the most common types of abuse against older adult women [42,61,62]. The apparently high prevalence of psychological abuse against older adult women may be due to a combination of factors, including their vulnerability and dependence [63]. Although psychological abuse can manifest itself with considerable frequency, its detection can be challenging due to its subjective and less visible nature compared to other forms of abuse [64]. However, while this type of abuse may be prevalent, its identification and addressing may be hampered by a lack of awareness or training among health professionals, including social workers [65]. In addition, older adult women may face additional barriers to reporting abuse due to psychological, social, geographical and cultural factors, contributing to the perpetuation of underreporting [28]. According to the 2019 Macro-Survey on Violence Against Women [66], in Spain, abused women ages ≥ 65 make less use of formal support services (such as psychological or psychiatric care) for physical, sexual or emotional abuse than younger women. In this sense, even though our study identified a 36.4% prevalence of psychological abuse—higher than any other subtype—its true magnitude may be underestimated.

Regarding neglect, the prevalence identified in the present study is significantly lower than that reported in previous studies [67,68]. This discrepancy may be related to the strong culture of family protection and intergenerational care within Spanish society, where the fulfilment of basic needs is widely regarded as a familial duty [23].

In our results we also found statistically significant relationships ($p < 0.05$) between abuse in any form, or any of its different types, and age, educational level, marital status, dependency status, use of social services, feelings of loneliness and general self-perceived health levels.

Firstly, we found a significant association between age and the prevalence of any abuse, and, specifically, psychological abuse. Women in the 60–69 and 70–79 age groups showed a higher prevalence of general abuse compared to those aged 80 years and older. This trend of decreasing abuse with increasing age among older adults is consistent with the findings of some previous studies (such as Dong et al. [69]; Mouton et al. [70]), although it contradicts with those of others (Burnes et al. [71]; Wu et al. [72]). Rather than a reduction in frequency, qualitative evidence seems to suggest that abuse transforms from physical forms to more subtle or "invisible" ones, possibly due to changes in the balance of power in families, which can make its identification difficult [73]. This could explain why the age variable was significantly related to psychological abuse in our study, highlighting the possible evolutionary nature of abuse in old age.

The analyses also detected a significant association between educational level and physical abuse, being more frequent among women with a medium-high education level than among women with low ones. Although without significant data, this was constant for the rest of the types of abuse analyzed. This may be because women with a medium-high educational level, regardless of their age, seem to be more prepared in several aspects that could influence their propensity to identify abusive behaviors [74], such as a greater knowledge of individual rights and laws related to violence, although this does not always translate into significant changes in the incidence of violence [75]. These findings are also relevant to inspire parallel care and prevention strategies, such as psychoeducational support groups [76] in women with low educational levels who suffer from violence.

Marital status was also revealed as a significant factor, as widowed, divorced, single, and separated older adult women showed a higher prevalence of abuse compared to married women. Additionally, those who experienced feelings of loneliness suffered greater physical, psychological, economic, and sexual abuse. These findings align with existing literature, which indicates that the social networks of women in violent relationships are often limited, resulting in less support [77]. Furthermore, loneliness appears as a central experience for older adult women who are victims of abuse, influencing their identity and relationships, and intensifies over time in abusive environments [78]. More evidence is required specifically aimed at the study of older adult women in abusive situations.

Interestingly, in our study women who did not report a dependency situation experienced higher levels of abuse, especially sexual abuse and neglect. Although the risk of abuse by caregivers of dependent older adult people has been documented [79,80], in Spain, to our knowledge, there is no recent research that addresses this situation in relation to professional caregivers of the Home Help Service, in many cases in charge of their care at home. Recently, these professionals have been receiving training for the prevention and detection of the abuse against older adult women [81,82], which could reduce the probability of exposure to situations of family abuse or IPV. Further research is required in this emerging area.

The use of social services by older adult women appeared linked to an increase in economic abuse. This finding may suggest that women who use these services could be experiencing greater vulnerability, making them more prone to different forms of abuse, especially economic. Although social workers in public social services are alert to economic abuse, and attempt to economically empower women, in many cases ongoing abuse is not addressed or monitored [83]. This suggests the need for greater professional training in order to effectively combat the structural factors related to this economic abuse and be able to simultaneously improve individual interventions that help older adult people protect themselves [84]. It is essential that social workers acquire good training to effectively address abuse in close relationships, given its global prevalence and significant impact on people's lives [85].

## Limitations

This study has several limitations that must be recognized and taken into account when interpreting its results. Firstly, the sample has limited representativeness, as it is composed mainly of older adult women from Andalusia, which restricts the generalizability of the results to other regions or populations. In addition, the overall sample size is relatively small, which further constrains the external validity of the study. Related to this, some variables include a low number of cases,

potentially reducing statistical power and limiting the ability to detect significant patterns or associations. Furthermore, the non-probability purposive sampling strategy—conducted through women's associations and social services—may introduce selection bias, as participants who engage with these institutional networks may differ systematically from the broader population of older adult women in terms of social connectedness, awareness of abuse, or prior contact with support services. As a result, the prevalence estimates reported in this study may not be directly generalisable to the general population of older adult women in Eastern Andalusia, Spain, or other settings, and should be interpreted as indicative findings from a purposively recruited community sample.

Another important limitation is the lack of differentiation of the perpetrator's profile in cases of abuse, which makes it difficult to accurately distinguish IPV from other types of abuse. Previous research geographically and culturally close to this study has reported that most cases of elder abuse were perpetrated by men in the immediate family environment; specifically, husbands and sons [86]. Although it is mostly assumed that abusers may be husbands in IPV relationships, or other family members (who may also be informal caregivers), this lack of distinction may affect the accuracy of the results and understanding the dynamics of abuse.

Third, the study focuses exclusively on older adult women, which excludes the experience of older adult men, who may also be victims of abuse. At the national level, it has been pointed out that even older adult men may suffer from greater neglect than older adult women [87]. This limitation could undermine a complete understanding of the phenomenon of elder abuse by leaving out an important perspective. However, it is relevant to highlight that there are few samples of studies in the field that focus solely on older adult women. Although older adult women tend to predominate in samples that include both older adult women and men, specifically exploring the group of older adult women offers a unique opportunity to delve into this topic in a more detailed and comprehensive manner.

Forth, variables were identified where no significant data were detected, such as place of residence, income level, housing situation, cohabitation status and participation in community activities. These findings suggest the need to observe the results related to these variables with caution, since they may not be representative or provide useful information to understand the phenomenon of abuse against older adult women in its entirety in this specific context.

Another limitation is related to the use of a backward stepwise (likelihood ratio) logistic regression approach. Although widely applied, this method may exclude variables that are not statistically significant despite their theoretical or practical relevance. To mitigate this risk, the interpretation of the findings considered not only statistical significance but also contextual and substantive relevance. Additionally, backward stepwise selection is susceptible to model instability in small-to-moderate samples, may capitalise on chance associations, and does not guarantee that the selected model is the theoretically most appropriate. Future studies with larger samples are encouraged to employ block-entry or theory-driven regression strategies to complement or validate these exploratory findings. To mitigate this risk, the interpretation of the findings considered not only statistical significance but also contextual and substantive relevance. Finally, previous studies indicate that cultural and social factors also have a significant influence. This could explain the disparity in findings reported in studies conducted in different sociocultural or temporal contexts on this problem, since in some circumstances abuse may be normalized or underreported, while in others there are better mechanisms for awareness and reporting. Likewise, temporal changes, such as the increase in awareness campaigns over time, could explain the different rates at different times [88]. However, this study does not address more sensitive analyses that would allow for a deeper understanding of sociocultural patterns with a causal impact on the phenomenon studied. A further measurement limitation concerns the internal consistency of the GMS in this sample. The moderate McDonald's omega ($\omega = .69$) is expected for a heterogeneous multidimensional scale covering conceptually distinct abuse subtypes; however, it indicates that items within the scale do not covary as strongly as in unidimensional constructs. In addition, because only binary subtype-level indicators were available in the dataset, it was not possible to compute reliability estimates for each individual abuse subtype, which would have required item-level data. Future studies should retain item-level GMS responses to

enable more granular psychometric evaluation, including omega estimates per subtype and confirmatory factor analysis of the scale's dimensional structure in Spanish community samples.

## Implications

Despite the limitations recognized, we believe that this study has the potential to generate various implications, mainly at the regional level, and may also provide inspiration for the field at the national and international levels.

These findings highlight the importance of professionals in the social and health field, especially social workers and geriatric health care providers, being trained to identify and address this issue. Greater awareness is needed about psychological abuse, which may be underestimated but is certainly prevalent. Additionally, the findings suggest that older adult women who use social services may be particularly vulnerable to economic abuse. Social services must improve their detection of and support for these women in order to reduce their vulnerability. Within these intervention settings, social work professionals should pay close attention to key vulnerability factors such as loneliness and dependency when assessing risk, by developing specific assessment instruments or improving existing ones. A holistic and sensitive approach is needed to address the complex family, social and cultural dynamics that surround this phenomenon.

For this purpose, it is of utmost importance to develop an evidence-based policy strategy that addresses the issue in a comprehensive and integrated manner. Specifically, there is a need for public health and social policies that address the abuse of older adult women, focusing on prevention, early detection and effective intervention. It is crucial that social policies address the socioeconomic and gender disparities that contribute to elder abuse. Specific strategies are needed to protect older adult women with lower levels of resources and social support. Social service policies must ensure that older adult women receive the support necessary to maintain their autonomy and security at home. This also includes the development of policies that provide greater training for professionals so that they can recognise and respond to it effectively and discreetly. In short, understanding the risk factors and sociodemographic characteristics that increase the vulnerability of women in the specific context is crucial for guiding professional intervention, as well as for promoting the development of evidence-based preventive and identification programs.

Longitudinal studies are needed to better understand the dynamics of the abuse against older adult women and how these may change over time. Future research should assess the effectiveness of preventive interventions aimed at reducing such abuse, focusing on the risk factors identified, such as loneliness and a lack of resources. Furthermore, the use of mixed-methods approaches would overcome some of the identified limitations, such as underdiagnosis, thereby increasing the accuracy and applicability of the findings In this regard, a qualitative approach can provide a deeper understanding of the experiences of abused older adult women. Additionally, it would be pertinent to conduct studies that incorporate the perspectives of professionals who work with older adult women in the context of abuse would be highly relevant. It would allow to analyze how this is perceived and addressed both from the perspective of social service professionals and the abused women themselves. This could inform best practices and policies in the field of social work.

## Conclusions

By evaluating the prevalence and factors associated with the abuse against older adult women in a region of Eastern Andalusia, this study contributes to a deeper understanding of a socially relevant problem that remains understudied in the specialised literature. The results reveal a high prevalence of abuse, with almost half of older adult women surveyed reporting having experienced some type of abuse in the past 12 months. Specifically, psychological abuse was the most commonly reported type, followed by physical and economic. The findings highlight the complexity of abuse in this population, with sociodemographic and lifestyle factors playing significant roles.

Age emerged as a risk factor, with women ages 60–79 showing a higher prevalence of abuse compared to those over 80. This trend may indicate a transformation in the nature of abuse as women age, with more subtle or "invisible" forms of

abuse potentially emerging at older ages. Furthermore, educational level and marital status were associated with different forms of abuse, suggesting the importance of addressing educational and social disparities to prevent abuse against this population group.

Loneliness also appeared as a significant factor, being associated with a higher incidence of general abuse. This underscores the importance of social networks and emotional support to prevent the abuse of older adult women. Additionally, the use of social services was associated with an increased risk of economic abuse, highlighting the need to improve resources and training to address economic vulnerabilities in this population. It is crucial that health professionals and social workers be trained to detect and address it, especially considering its underreporting and barriers to reporting that may exist. In addition, greater investment is needed in prevention and public education programs to address the attitudes and social norms that perpetuate and obscure this phenomenon.

## Acknowledgments

We would like to express our sincere gratitude to all the women participants in this study, whose courage and trust have been fundamental. We hope that the findings obtained will contribute to improving their situation and benefiting future generations of women. Their contribution has been invaluable to the success of this study. Specifically, we would like to mention the following collaborating entities: Asociación de Mujeres del Donadío, Asociación de Amas de Casa Moreira, Asociación de Mujeres la Aldea de Santa Eulalia, Asociación Mujeres Violeta, Asociación Mujeres Viudas María de Molina, Asociación Nacer y Mamar, Asociación de Mujeres Encajeras Ciudad de Úbeda, Asociación Mujeres Vecinales Libertad, Asociación Mujeres Valientes, Federación de Asociaciones Vecinales "La Loma de Úbeda," Cáritas, Cruz Roja, Centro de Participación Activa and the Centro de Día Residencia el Carmen.

## Author contributions

**Conceptualization:** Adrián Jesús Ricoy-Cano, María Dolores Muñoz-de-Dios.

**Data curation:** Adrián Jesús Ricoy-Cano, María Dolores Muñoz-de-Dios, Marta García-Domingo.

**Formal analysis:** Adrián Jesús Ricoy-Cano, María Dolores Muñoz-de-Dios.

**Funding acquisition:** Yolanda María de la Fuente-Robles, María Dolores Muñoz-de-Dios.

**Investigation:** Yolanda María de la Fuente-Robles, Adrián Jesús Ricoy-Cano, María Dolores Muñoz-de-Dios, Marta García-Domingo.

**Methodology:** Adrián Jesús Ricoy-Cano.

**Project administration:** Yolanda María de la Fuente-Robles, María Dolores Muñoz-de-Dios.

**Software:** Adrián Jesús Ricoy-Cano, Marta García-Domingo.

**Supervision:** Yolanda María de la Fuente-Robles, Adrián Jesús Ricoy-Cano, María Dolores Muñoz-de-Dios.

**Validation:** Yolanda María de la Fuente-Robles.

**Visualization:** Adrián Jesús Ricoy-Cano, Marta García-Domingo.

**Writing – original draft:** Adrián Jesús Ricoy-Cano, María Dolores Muñoz-de-Dios, Marta García-Domingo.

**Writing – review & editing:** Yolanda María de la Fuente-Robles, Adrián Jesús Ricoy-Cano.

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
