## [Decision Letter · Decision Letter 0]

4 Jun 2025

PONE-D-24-58875The prevalence of the mistreatment of older women and associated risk factors in an area of Eastern Andalusia, Spain: A cross-sectional, community-based studyPLOS ONE

Dear Dr. Ricoy-Cano,

Thank you for submitting your manuscript to PLOS ONE. After careful consideration, we feel that it has merit but does not fully meet PLOS ONE’s publication criteria as it currently stands. Therefore, we invite you to submit a revised version of the manuscript that addresses the points raised during the review process.

The manuscript describes a technically sound piece of scientific research with data that supports the conclusions.

We look forward to receiving your revised manuscript.

Kind regards,

Doris V. Ortega-Altamirano, PhD

Academic Editor

PLOS ONE

Journal Requirements:

This work was supported by the EXCMO. Ayuntamiento de Úbeda under Grant number 2022168. Project title: Detection and analysis of situations of violence in older women in Úbeda and its surrounding areas. The author(s) also reported receiving the following financial support for authorship of this article: One author (Adrián Jesús Ricoy-Cano) was funded by the European Union through the “NextGenerationEU” program, as part of the “Grants for the Requalification of the System” program. Spanish University for 2021-2023 under the “MARGARITA SALAS” modality.

5. Please note that your Data Availability Statement is currently missing the DOI/accession number of each dataset OR a direct link to access each database. If your manuscript is accepted for publication, you will be asked to provide these details on a very short timeline. We therefore suggest that you provide this information now, though we will not hold up the peer review process if you are unable.

6. Please provide a complete Data Availability Statement in the submission form, ensuring you include all necessary access information or a reason for why you are unable to make your data freely accessible. If your research concerns only data provided within your submission, please write "All data are in the manuscript and/or supporting information files" as your Data Availability Statement.

7. When completing the data availability statement of the submission form, you indicated that you will make your data available on acceptance. We strongly recommend all authors decide on a data sharing plan before acceptance, as the process can be lengthy and hold up publication timelines. Please note that, though access restrictions are acceptable now, your entire data will need to be made freely accessible if your manuscript is accepted for publication. This policy applies to all data except where public deposition would breach compliance with the protocol approved by your research ethics board. If you are unable to adhere to our open data policy, please kindly revise your statement to explain your reasoning and we will seek the editor's input on an exemption. Please be assured that, once you have provided your new statement, the assessment of your exemption will not hold up the peer review process.

Additional Editor Comments:

The authors should make it clear in the introduction why it's crucial to document the problem of violence and abuse against older adults.

It is necessary for the authors to clarify: Even if the terms: violence," "mistreatment," and "abuse" are related in a very close way, it is necessary to deepen them conceptually and specify in the manuscript to which they refer. Consider that they are not synonymous

For the title, there is no need to cut it to 20 words.

Unify in the manuscript, where possible, the term: older adults.

Emphasize how the study contributes to broadening knowledge of the phenomenon of abuse to older adults.

Clarifying in methods: the context of older adult women being studied. And, where appropriate, separate; women in their homes by type of family and women in specialized institutions for elderly care.

Clarify the type of sampling performed. There are two opposing views: snowball and non-probabilistic sample. What was actually done?

Reviewers' comments:

Reviewer's Responses to Questions

**Comments to the Author**

1. Is the manuscript technically sound, and do the data support the conclusions?

Reviewer #1: Partly

Reviewer #2: Yes

2. Has the statistical analysis been performed appropriately and rigorously? 

Reviewer #1: Yes

Reviewer #2: Yes

3. Have the authors made all data underlying the findings in their manuscript fully available?

Reviewer #1: Yes

Reviewer #2: Yes

4. Is the manuscript presented in an intelligible fashion and written in standard English?

Reviewer #1: Yes

Reviewer #2: Yes

5. Review Comments to the Author

Reviewer #1: The prevalence of the mistreatment of older women and associated risk factors in an

area of Eastern Andalusia, Spain: A cross-sectional, community-based study

All my comments are categorized as major, minor and recommendation. You may or may not consider addressing the comments that are only meant for recommendation. I hope my comments will help increase the quality of your work.

Title

<minor> Per PLoS guidelines, you should include a Short Title as Running Head.

<recommendation> Although your full title is under suggested 250 characters, I suggest keeping it under 20 words.

Abstract

Kindly note that my comments on the abstract may not be repeated in the main text, but they are intended to apply wherever relevant throughout the manuscript.

<major> The use of ageist language such as “elderly,” “aged,” “seniors” to describe older adults is highly discouraged in scientific literature. Please use other aging friendly language such as “older adults,” “older persons” throughout the manuscript. However, since “elder abuse” is a legal term, this term is accepted for its global legal implication. On the other hand, “elder” is considered somewhat accepted; however, I still discourage using this term.

<minor> While it is often stated that research on the mistreatment of older women in Spain is scarce, such claims can sometimes be overly general. Since your study is only about prevalence and associated factors, please reconfirm this assertion of research gap under this topic. And also, kindly discuss the strength, if you have not, under the “Strengths and Limitations” your study provides to fill up this research gap on older women in Spain.

<recommendation> The words like “violence,” “mistreatment,” and “abuse” may be closely related, but they are not the same concepts. Please use one of the terms that your study is primarily about and be consistent throughout the manuscript. You may describe all three terms briefly in the introduction and state which of these terms your study is specifically about. This will enhance clarity on what your study is specifically about.

<recommendation> You specified that your study is among non-institutionalized women. However, I find your conclusions for the abstract more relevant to abuse among institutionalized women, and thus you may require making it more relevant or distinct to address issues of abuse in community settings, including specific recommendations to address the issues of abuse in household settings.

Introduction

Your introduction is well-written covering major aspects of providing a background, highlighting the existing research and potential research gap. However, I have a few additional suggestions:

<major> I realized that your study is primarily about abuse in domestic settings. Please make sure you do not mix up the concepts of violence against older women with abuse against older women although these two concepts intersect; and please make sure that your writings and cited literature are primarily about abuse among older women, not about violence against women. Violence against women is another broad topic of research studied widely in Spain.

<major> International readers might not know what type of society exist in Spain. For example, is it an individualistic or collectivist society in Spain? Are families in Spain multigenerational or nuclear? Is the older care provided by family members or other informal caregivers or by assisted living professionals? These background information help readers to understand the settings related to older care in Spain.

<major> Although you discussed very briefly that there are strong regulations to protect abuse against older women, you might consider adding a separate short paragraph for global and local readers to learn what specific legal frameworks are tailored to protect older women against abuse in Spain. And, based on existing literature, I suggest you provide some justification behind higher prevalence of elder abuse among women in Spain, despite robust laws and policies against elder abuse.

<recommendation> Please add a conceptual framework to enhance clarity.

<recommendation> If you can, it is better to integrate a relevant gerontological theory to strengthen your study. Please discuss one suitable relevant gerontological theory in the introduction section and elaborate your findings in support of the existing theoretical framework.

Materials and methods

Study design, sites and participants

<major> You need to clarify the sampling approach used in your study. Was the design based on probability sampling, or did it follow a purposive or non-probability approach? The first paragraph suggests characteristics of snowball sampling, but this is not explicitly stated. On the contrary, the second paragraph suggests a random sampling. If you have used random sampling, please detail out the process. Also, please provide a rationale behind your sample size. Clear identification of the sampling method is essential for understanding the representativeness and limitations of your findings.

<minor> You have hinted a little. However, I am curious to learn whether the study was undertaken within their household or outside of the household. If it was taken in the household, what measures were taken to avoid the interruption from other family members given the sensitivity of the topic? It will be helpful to add whether or not proxies were allowed. Also, how was the data collected? Did you use paper surveys or tablet questionnaire? If paper surveys were used, how did you digitize the data. If tablet version was used, please elaborate a little on how you designed the software and performed data management.

Measures

Geriatric Mistreatment Scale

<recommendation> Please specify how you checked the internal consistency of the Geriatric Mistreatment Scale in your study. I am asking this question because if you have used Cronbach alpha, although widely used it is not considered a robust measure to assess internal consistency. If you want to, you may consider using alternative measures such as split half or McDonald’s Omega instead of using Cronbach alpha to test reliability.

Socio-demographic information

<major> The information on socio-demographic information lacks clarity. Were the variables recoded during the data analysis? Or were the variables collected in the way those were explained. If it is the former case, please make sure to detail out how the variables were recoded. Your explanation of the categories is not sufficient. For example, how did you determine whether the place was urban or rural. What standard was used? Likewise, what did “enough” vs “not enough” income level mean? These are some examples. Please add clarity to all other variables.

Statistical analyses

<minor> How was the data imported into SPSS environment? If paper-based survey was used, what were the measures taken to minimize the data-entry bias during the digitization process?

<minor> Please clarify the type of logistic regression you used to assess abuse. Is it adjusted multivariable logistic regression?

<major> Did you check for multicollinearity issues? If not, please do. Also, consider performing advanced diagnostics to check for influential observations, and conduct the treatment if necessary. Please report the value under AUROC, preferably concordance statistics for logistic regression, and then cite relevant methodological paper.

<major> Regression often washes out the variables with missing data. So, please clarify if you used any imputation process to address missing data? Or, please clarify what your final analytic sample was.

Ethical considerations

Please add a statement describing that participant had the right to refuse to response to any question, terminate the interview at any time without penalty, and were also given the option to withdraw after the interview.

Results

Maltreatment and socio-demographic/lifestyle variables

<major> Although you have several types of elder abuse enlisted, you have not defined what each of these types are under the methods section. Please, provide an operational definition to each of these subtypes, and how you measured each type and overall abuse.

<major> Please check your results for Table 2 and make sure that all of the bivariate relationships were assessed using Chi-square test, as some cell counts show potential implication of alternative method like Fisher Exact Test.

Risk factors for abuse of older women

<major> Please clarify what type of logistic regression did you use? Also, indicate your cut-off point for p-values and level of confidence intervals.

While you have mentioned using the backward elimination method, it would be helpful to briefly explain this technique for readers who may not be familiar with it. Specifically, you could describe how variables are removed based on statistical significance and how this impacts model interpretation. This description should be integrated under the methods section.

Could you also clarify whether you used a stepwise regression for backward elimination? If stepwise regression was applied, I recommend revisiting the model selection process. These techniques, although commonly used, have known limitations. They can sometimes exclude variables that may not be statistically significant but have important theoretical or practical implications. This description should be integrated under the methods section.

Consider discussing the rationale behind your modeling choices, especially in the context of your research question. Including variables with substantive relevance—even if they're not statistically significant—can enhance the interpretability and applicability of your findings. Also, kindly note that p-values are not definitive indicators of variable importance and I suggest not to solely rely on p-values. Those closer to the cut-off point of 0.05 might still provide some evidence regarding the association. Therefore, you may consider describing the variables that are not statistically significant but approach significance.

<minor> You may remove the column indicating β values in Table 3, since OR=e^β in logistic regression can be easily derived.

<major> Please consider revising the interpretation for the log-odds for the first one or two results in each paragraph. This helps the reader to interpret and understand the results correctly. For example, include at least one or two interpretations in the general way logistic regressions are interpreted, including the variables you controlled . For example, after adjusting for marital status and general health, participants aged 60–69 had more than twice the odds of experiencing overall abuse compared to those aged 80 or older (OR = 2.36; 95% CI: 1.11–4.99). Similarly, those aged 70–79 had 187% higher the odds (OR = 2.87; 95% CI: 1.40–5.87) of facing overall abuse than those 80+ years after holding other variables in the model constant. Also, please report CI instead of p-values along with OR in your logistic regression interpretations. Confidence intervals are preferred more than the p-values. Please also restructure your table 3 to show what the reference category for each variable was.

Discussion

<minor> In the first paragraph, instead of reporting what sociodemographic variables were investigated, kindly report what variables were found to establish significant association with abuse.

<major> Neglect and psychological abuse are two most prevalent types of abuse. It was interesting to see such a low prevalence in your study. Kindly provide a strong justification behind such low prevalence. Also, please balance your discussion discussing both higher and lower ends of prevalence of neglect in literature. This applies to all other sections wherever applicable.

<major> To balance your discussion section, also discuss some of the variables that were close to significance, yet not significant and provide the justification in light of existing literature.

<recommendation> If you integrate a theory in the introduction section, kindly discuss your findings supporting or refuting the theory.

<recommendation> Please reduce the length of implications. You may not require three separate subheadings.</recommendation></recommendation></major></major></minor></major></minor></major></major></major></major></major></minor></minor></major></recommendation></minor></major></recommendation></recommendation></major></major></major></recommendation></recommendation></minor></major></recommendation></minor>

Reviewer #2: I would like to express my sincere gratitude to the editor for the opportunity to review the study titled "The prevalence of the mistreatment of older women and associated risk factors in an area of Eastern Andalusia, Spain: A cross-sectional, community-based study." I appreciate the effort into this research and look forward to providing constructive feedback to enhance its quality and impact. I have outlined several comments and suggestions below that aim to strengthen the overall quality and clarity of the manuscript:

1. Some parts are overly verbose. Aim for more concise sentences to maintain reader engagement.

2. The literature review could be strengthened by including more recent studies or contrasting findings to highlight gaps in existing research.

3. The sample size (209 participants) may limit the generalizability of the findings. Write about sample size.

4. More detail on the Geriatric Maltreatment Scale's validation process would enhance credibility. How was it adapted for this study?

5. Some claims lack sufficient backing from the data presented. Ensure that all assertions are supported by the results in discussion.

6. The implications for practice and policy could be more explicitly stated. What specific actions should be taken based on the findings?

7. The discussion of limitations could be expanded. Consider discussing the potential impact of cultural factors on the findings.

8. It could reiterate the importance of addressing the identified issues in a more impactful manner.

9. Suggest future research directions more explicitly to guide subsequent studies.

10. The manuscript is generally well-written, but proofreading for grammatical errors and clarity is recommended.

6. PLOS authors have the option to publish the peer review history of their article (what does this mean?). If published, this will include your full peer review and any attached files.

Reviewer #1: **Yes:** Aman Shrestha

Reviewer #2: No

---

## [Author Response · Author response to Decision Letter 1]

15 Dec 2025

RESPONSE TO THE EDITOR AND REVIEWERS

Manuscript ID PONE-D-24-58875

We would like to express our sincere appreciation to the editors and reviewers of PLOS ONE for their careful review and constructive feedback. We have thoroughly addressed all the comments provided, and we believe that the manuscript has been significantly strengthened as a result.

Below, we provide a detailed, point-by-point response to each of the reviewers’ suggestions and concerns.

RESPONSE TO THE EDITOR'S COMMENTS

Additional Editor Comments:

Editor (1.1):

The authors should make it clear in the introduction why it's crucial to document the problem of violence and abuse against older adults. It is necessary for the authors to clarify: Even if the terms: violence," "mistreatment," and "abuse" are related in a very close way, it is necessary to deepen them conceptually and specify in the manuscript to which they refer. Consider that they are not synonymous.

Authors (1.1): We appreciate the editor’s insightful observation, particularly given the complexity and cross-national variability surrounding the terminology of violence, mistreatment, and abuse in later life. Following this recommendation, we have strengthened the introduction by explicitly justifying the importance of documenting abuse against older adults’ women and by clarifying the conceptual distinctions between violence, mistreatment, and abuse. We now specify more clearly that the manuscript focuses on abuse within domestic and care-related contexts, acknowledging its conceptual boundaries while recognising the broader continuum of violence affecting older adult women. These revisions aim to enhance conceptual precision and align the terminology with current international scholarship in the field.

Editor (1.2): For the title, there is no need to cut it to 20 words. Unify in the manuscript, where possible, the term: older adults. Emphasize how the study contributes to broadening knowledge of the phenomenon of abuse to older adults.

Authors (1.2): We thank the Editor for these helpful indications. In response, we have slightly revised the title to improve its precision while maintaining clarity and coherence with the study’s aims. We have also standardised the terminology throughout the manuscript, using the term older adults consistently where appropriate. In addition, we have strengthened the Introduction and Discussion to more explicitly highlight how the study contributes to broadening current knowledge on abuse against older adults. These changes have now been incorporated into the revised manuscript.

Editor (1.3): Clarifying in methods: the context of older adult women being studied. And, where appropriate, separate; women in their homes by type of family and women in specialized institutions for elderly care. Clarify the type of sampling performed. There are two opposing views: snowball and non-probabilistic sample. What was actually done?

Authors (1.3): In response, we have clarified the study context by explicitly stating that all participants were community-dwelling older adult women living in their own homes or with family members, and that no women residing in specialised long-term care institutions were included. We have also specified the sampling approach, indicating that the study used a non-probability, purposive sampling strategy, rather than snowball sampling. These clarifications have now been incorporated into the Methods section to ensure full transparency and alignment with the study design.

RESPONSE TO REVIEWERS’ COMMENTS

REVIEWER 1

5. Review Comments to the Author

Reviewer #1: The prevalence of the mistreatment of older women and associated risk factors in an area of Eastern Andalusia, Spain: A cross-sectional, community-based study

All my comments are categorized as major, minor and recommendation. You may or may not consider addressing the comments that are only meant for recommendation. I hope my comments will help increase the quality of your work.

Authors (1.0): Dear reviewer, thank you very much for the time devoted to reviewing our manuscript and for your thoughtful suggestions for its improvement. All modifications made in response to your comments can be clearly viewed in the version of the manuscript with track changes enabled, as required by the journal’s editorial guidelines.

Reviewer (1.1):

Title

Per PLoS guidelines, you should include a Short Title as Running Head.

Although your full title is under suggested 250 characters, I suggest keeping it under 20 words.

Authors (1.1): Thank you for your suggestion regarding the manuscript title. Following your recommendation, the title has been revised and now contains fewer than 20 words, while preserving clarity and relevance. Additionally, a short title has been provided in accordance with the journal’s editorial guidelines.

Reviewer (1.2):

Abstract

Kindly note that my comments on the abstract may not be repeated in the main text, but they are intended to apply wherever relevant throughout the manuscript.

The use of ageist language such as “elderly,” “aged,” “seniors” to describe older adults is highly discouraged in scientific literature. Please use other aging friendly language such as “older adults,” “older persons” throughout the manuscript. However, since “elder abuse” is a legal term, this term is accepted for its global legal implication. On the other hand, “elder” is considered somewhat accepted; however, I still discourage using this term.

While it is often stated that research on the mistreatment of older women in Spain is scarce, such claims can sometimes be overly general. Since your study is only about prevalence and associated factors, please reconfirm this assertion of research gap under this topic. And also, kindly discuss the strength, if you have not, under the “Strengths and Limitations” your study provides to fill up this research gap on older women in Spain.

The words like “violence,” “mistreatment,” and “abuse” may be closely related, but they are not the same concepts. Please use one of the terms that your study is primarily about and be consistent throughout the manuscript. You may describe all three terms briefly in the introduction and state which of these terms your study is specifically about. This will enhance clarity on what your study is specifically about.

You specified that your study is among non-institutionalized women. However, I find your conclusions for the abstract more relevant to abuse among institutionalized women, and thus you may require making it more relevant or distinct to address issues of abuse in community settings, including specific recommendations to address the issues of abuse in household settings.

Authors (1.2): We sincerely thank the reviewer for these valuable observations. In response to the suggestions:

We have made all the requested terminological adjustments both in the abstract and consistently throughout the manuscript. Specifically, we have ensured the use of the expression “abuse against older adult women” as the central and most appropriate terminology, avoiding ageist terms such as “elderly,” “seniors,” or “aged,” and favouring people-first, aging-friendly language. The only exception is the use of “elder abuse” in reference to legal or internationally standardised terminology, as suggested.

Additionally, we have emphasised throughout the text that the study is focused exclusively on non-institutionalised older adult women, and have revised both the abstract and conclusion to make this distinction clearer. Specific references to abuse in household and community settings have been included to ensure that the conclusions and recommendations align with the actual scope and population of the study.

We have also revised the Introduction to clarify the conceptual distinctions between violence, mistreatment, and abuse, and to explain the specific focus and terminology adopted in our study to enhance conceptual clarity.

Finally, Strengths and Limitations section has been expanded to better frame the contribution of this study within the broader research context on older adult women in Spain, avoiding overgeneralisations while underscoring the study’s relevance and specificity. We modified the conclusions of the abstract to be more aligned with the findings and scope of this study.

Reviewer (1.3):

Introduction

Your introduction is well-written covering major aspects of providing a background, highlighting the existing research and potential research gap.

However, I have a few additional suggestions:

I realized that your study is primarily about abuse in domestic settings. Please make sure you do not mix up the concepts of violence against older women with abuse against older women although these two concepts intersect; and please make sure that your writings and cited literature are primarily about abuse among older women, not about violence against women. Violence against women is another broad topic of research studied widely in Spain.

Authors (1.3): Thank you for this observation. We have revised the introduction to make a clearer and more consistent distinction between abuse against older adult women and the broader concept of violence against women. We ensured that the definitions, terminology, and cited literature focus specifically on elder abuse in domestic settings, and we now explicitly clarify that our study adopts the narrower and more precise concept of abuse, avoiding conceptual overlap with general VAW research.

Reviewer (1.4):

International readers might not know what type of society exist in Spain. For example, is it an individualistic or collectivist society in Spain? Are families in Spain multigenerational or nuclear? Is the older care provided by family members or other informal caregivers or by assisted living professionals? These background information help readers to understand the settings related to older care in Spain.

Authors (1.4): We thank the reviewer for this valuable suggestion. In response, we have strengthened the introduction by explicitly contextualising Spain as a familistic and predominantly collectivist welfare setting, characterised by strong intergenerational ties and a care model largely sustained by family members—especially women. This information has been incorporated to clarify that elder care in Spain is primarily provided within the household, usually by informal caregivers rather than formal services, thereby offering international readers a clearer understanding of the sociocultural and care structures relevant to interpreting the study's findings.

Reviewer (1.5):

Although you discussed very briefly that there are strong regulations to protect abuse against older women, you might consider adding a separate short paragraph for global and local readers to learn what specific legal frameworks are tailored to protect older women against abuse in Spain. And, based on existing literature, I suggest you provide some justification behind higher prevalence of elder abuse among women in Spain, despite robust laws and policies against elder abuse.

Authors (1.5): We appreciate this valuable suggestion. In response, we have added a clearer and more explicit description of the Spanish legal context in the introduction, specifying the main regulatory frameworks relevant to the protection of older adult women—namely, Organic Law 1/2004 on gender-based violence and the provisions of the Spanish Criminal Code (Art. 173 and related articles). We also clarified that Spain lacks a specific legal definition of elder abuse, which results in fragmented protection. Additionally, we incorporated a brief explanation, supported by existing literature, discussing how the persistence of higher prevalence rates among older adult women may be related to sociocultural factors (e.g., familism, underreporting, dependency, and invisibility of psychological abuse).

Reviewer (1.6):

Please add a conceptual framework to enhance clarity. If you can, it is better to integrate a relevant gerontological theory to strengthen your study. Please discuss one suitable relevant gerontological theory in the introduction section and elaborate your findings in support of the existing theoretical framework.

Authors (1.6): A conceptual framework has now been explicitly incorporated in the introduction by drawing on established etiological theories of elder abuse and integrating them within an ecological perspective. This framework clarifies how individual, relational, community, and sociocultural factors jointly shape risk among older adult women. In the discussion, we have briefly elaborated on how our findings align with this ecological configuration—particularly the interplay of vulnerabilities such as loneliness, dependency, and sociocultural conditions—thereby strengthening the theoretical coherence of the manuscript.

Reviewer (1.7):

Materials and methods

Study design, sites and participants

You need to clarify the sampling approach used in your study. Was the design based on probability sampling, or did it follow a purposive or non-probability approach? The first paragraph suggests characteristics of snowball sampling, but this is not explicitly stated. On the contrary, the second paragraph suggests a random sampling. If you have used random sampling, please detail out the process. Also, please provide a rationale behind your sample size. Clear identification of the sampling method is essential for understanding the representativeness and limitations of your findings.

Authors (1.7): Thank you again for your observation regarding the sampling method. We have revised the section to clearly indicate that a non-probability, purposive sampling approach was used, based on the networks and dissemination capacity of collaborating institutions. Additionally, we have justified the sample size according to feasibility criteria within the project’s time frame. These clarifications have been incorporated in the revised version of the manuscript.

Reviewer (1.8):

You have hinted a little. However, I am curious to learn whether the study was undertaken within their household or outside of the household. If it was taken in the household, what measures were taken to avoid the interruption from other family members given the sensitivity of the topic? It will be helpful to add whether or not proxies were allowed. Also, how was the data collected? Did you use paper surveys or tablet questionnaire? If paper surveys were used, how did you digitize the data. If tablet version was used, please elaborate a little on how you designed the software and performed data management.

Authors (1.8): We appreciate this detailed and thoughtful observation, which highlights key aspects of the data collection process that are indeed essential for fully understanding the fieldwork context. Following your suggestions, we have revised the section to provide a clearer description of where the interviews were conducted, how privacy and confidentiality were ensured (particularly in household settings), and to confirm that no proxy interviews were required. We also specify that data were collected using both paper and digital formats. In most cases, responses were directly entered into the online questionnaire (Google Forms) by members of the research team. In other cases, they were first recorded on paper and later transcribed into the digital version by trained support staff. These changes have been incorporated into the section Study design, sites and participants.

Reviewer (1.9):

Measures

Geriatric Mistreatment Scale

Please specify how you checked the internal consistency of the Geriatric Mistreatment Scale in your study. I am asking this question because if you have used Cronbach alpha, although widely used it is not considered a robust measure to assess internal consistency. If you want to, you may consider using alternative measures such as split half or McDonald’s Omega instead of using Cronbach alpha to test reliability.

Authors (1.9): Thank you for your observation. I

---

## [Decision Letter · Decision Letter 1]

9 Mar 2026

PONE-D-24-58875R1Prevalence and risk factors of abuse against older adult women: a cross-sectional community study in Eastern Andalusia, Spain

PLOS One

Dear Dr. Ricoy-Cano,

Thank you for submitting your manuscript to PLOS ONE. After careful consideration, we feel that it has merit but does not fully meet PLOS ONE’s publication criteria as it currently stands. Therefore, we invite you to submit a revised version of the manuscript that addresses the points raised during the review process.

We look forward to receiving your revised manuscript.

Kind regards,

Vandana Dabla, Ph.D.

Academic Editor

PLOS One

**Journal Requirements:**

**Additional Editor Comments:**

We acknowledge author efforts to address the reviwers comments, however, some methodological and interpretative issues require clarification, particularly regarding **sampling, measurement reliability and interpretation of the high prevalence estimate**.

1. The study used non-probability purposive sampling through institutional networks (women’s associations and social services). This approach may introduce selection bias, potentially affecting prevalence estimates. Thus, author must clarify the recruitment process in “more detail” and explicitly acknowledge the limitations for generalizability in the discussion section.

2. The reported prevalence of 49.3% abuse within 12 months is considerably higher than most international estimates. The study classifies a participant as abused if she answered “yes” to at least one of the 22 items in the Geriatric Mistreatment Scale, and this  approach inflates prevalence estimates. Thus, it becomes extremely vital to provide a stronger explanation in the discussion, noting that the estimate may reflect a) the screening nature of the instrument, b) classification of abuse based on any positive item out of 22 , c) and non-probability community sampling.

3. Internal consistency for the Geriatric Mistreatment Scale in this sample was moderate (ω ≈ 0.69). Author must report reliability estimates for each abuse subtype or provide additional justification for using the combined “any abuse” variable in the analysis.

4. The manuscript used backward stepwise logistic regression, which may lead to unstable models or exclusion of theoretically important variables. Thus, author must briefly justify the modeling strategy and discuss its limitations in the limitations section.

5. Ensure consistent use of the term “abuse against older adult women” throughout the manuscript.

6. Several socio-demographic variables (such as age, education, and income) were recoded into broad categories, which may reduce variability & statistical precision. The manuscript should briefly justify these recoding decisions and explain how the chosen categories align with previous research or contextual considerations.

7. Author may do minor language editing to remove repetitions.

Reviewers' comments:

Reviewer's Responses to Questions

**Comments to the Author**

1. If the authors have adequately addressed your comments raised in a previous round of review and you feel that this manuscript is now acceptable for publication, you may indicate that here to bypass the “Comments to the Author” section, enter your conflict of interest statement in the “Confidential to Editor” section, and submit your "Accept" recommendation.

Reviewer #1: All comments have been addressed

2. Is the manuscript technically sound, and do the data support the conclusions?

Reviewer #1: Yes

3. Has the statistical analysis been performed appropriately and rigorously? 

Reviewer #1: Yes

4. Have the authors made all data underlying the findings in their manuscript fully available?

Reviewer #1: Yes

5. Is the manuscript presented in an intelligible fashion and written in standard English?

Reviewer #1: Yes

6. Review Comments to the Author

Reviewer #1: I appreciate how the authors worked to improve the manuscript and address/integrate my comments/suggestions. Thank you for your sincere effort. I do notice that the internal reliability of the Geriatric Mistreatment Scale is not that great; but it is what it is based on the dataset you analyzed. If possible, please acknowledge this limitation in your manuscript and if you are able to justify briefly why you think your McDonald's omega value was only acceptable. Other than that, your revised version now has robust methods, clear presentation of results, and succint overview and discussion.

7. PLOS authors have the option to publish the peer review history of their article (what does this mean?). If published, this will include your full peer review and any attached files.

Reviewer #1: **Yes:** Aman Shrestha

---

## [Author Response · Author response to Decision Letter 2]

21 Apr 2026

RESPONSE LETTER TO EDITORS AND REVIEWERS

Manuscript ID PONE-D-24-58875R1

Title Prevalence and risk factors of abuse against older adult women: a cross-sectional community study in Eastern Andalusia, Spain

General Comment for Editors and Reviewers

We would like to express our sincere appreciation to the editors and reviewers of PLOS ONE for their careful review and constructive feedback. We have thoroughly addressed all the comments provided, and we believe that the manuscript has been significantly strengthened as a result.

Below, we provide a detailed, point-by-point response to each of the reviewers’ suggestions and concerns.

Additional Editor Comments:

We acknowledge author efforts to address the reviwers comments, however, some methodological and interpretative issues require clarification, particularly regarding sampling, measurement reliability and interpretation of the high prevalence estimate.

Editor’s Comment 1. The study used non-probability purposive sampling through institutional networks (women’s associations and social services). This approach may introduce selection bias, potentially affecting prevalence estimates. Thus, author must clarify the recruitment process in “more detail” and explicitly acknowledge the limitations for generalizability in the discussion section.

Authors’ Response 1.

We thank the Editor for this comment. We have expanded the description of the recruitment process in the “Materials and Methods” section to provide greater detail regarding the institutional networks involved, how contact with potential participants was facilitated, and the protective measures taken prior to data collection. Specifically, we now clarify that participants were recruited through women’s associations, neighbourhood organisations, and municipal social services (via the home help service provider), and that collaborating institutions received an advance letter presenting the research team and the purpose of the study.

In addition, we have explicitly acknowledged the limitations of non-probability purposive sampling in the Limitations section of the Discussion, noting that this approach may introduce selection bias and that the resulting estimates should not be generalised to the broader population of older adult women in Spain or beyond. These changes are marked in the revised manuscript.

Editor’s Comment 2. The reported prevalence of 49.3% abuse within 12 months is considerably higher than most international estimates. The study classifies a participant as abused if she answered “yes” to at least one of the 22 items in the Geriatric Mistreatment Scale, and this approach inflates prevalence estimates. Thus, it becomes extremely vital to provide a stronger explanation in the discussion, noting that the estimate may reflect a) the screening nature of the instrument, b) classification of abuse based on any positive item out of 22 , c) and non-probability community sampling.

Authors’ Response 2.

We fully agree with the Editor’s observation. We have strengthened the explanation of the high prevalence estimate in the Discussion section. The revised text now explicitly acknowledges that the figure of 49.3% should be interpreted in light of three methodological characteristics: (a) the GMS functions as a screening instrument designed to identify the possible presence of abuse, rather than to produce clinically confirmed case rates; (b) the classification criterion—any affirmative response to at least one of the 22 items—is broad by design and likely captures sub-threshold or borderline experiences that narrower instruments would not detect; and (c) the non-probability purposive sampling may have recruited women with greater exposure or awareness of abuse situations.

We have also added a direct comparison with other studies that used similar screening approaches and reported comparable or higher prevalence rates, contextualising our findings within the broader literature. These changes are clearly marked in the revised manuscript.

Editor’s Comment 3. Internal consistency for the Geriatric Mistreatment Scale in this sample was moderate (ω ≈ 0.69). Author must report reliability estimates for each abuse subtype or provide additional justification for using the combined “any abuse” variable in the analysis.

Authors’ Response 3.

We thank the Editor for this important methodological comment. We have addressed it through two complementary strategies.

First, regarding reliability estimates at the subtype level: the dataset available for analysis contains binary indicators reflecting whether each participant experienced a given subtype of abuse (yes/no), rather than item-level responses for each of the 22 individual items of the GMS. Because internal consistency indices such as McDonald’s omega or KR-20 require item-level data (i.e., the response to each individual item within a subtype), it is not possible to compute these statistics for each subtype directly from the available data. We have acknowledged this explicitly in the revised Measures section.

Second, to provide the additional justification requested for using the combined “any abuse” variable, we have added the following evidence-based arguments in the Measures section and in the Limitations section:

a) Convergent validity of the subtypes: The phi (φ) correlations between abuse subtypes—computed from the present dataset—demonstrate meaningful inter-subtype associations, particularly among psychological, economic, sexual, and neglect dimensions (e.g., φ = 0.47 for neglect–sexual; φ = 0.37 for psychological–sexual; φ = 0.33 for psychological–financial), consistent with the polyvictimisation literature on elder abuse. These associations support the conceptual coherence of a combined abuse indicator.

b) High co-occurrence of abuse forms: Of the 103 women who experienced any form of abuse (49.3%), 55.3% experienced two or more forms simultaneously (45.6% of all 103 abused women experienced multiple forms). This polyvictimisation pattern justifies the use of a composite “any abuse” variable as the primary outcome in logistic regression, since distinct subtypes frequently co-occur and are unlikely to operate as fully independent constructs.

c) Consistency with the GMS validation study and prior usage: The GMS was validated and has been used in prior research with the combined score as the primary outcome variable (Giraldo-Rodríguez & Rosas-Carrasco, 2013; Piña-Escudero et al., 2021; Vilar-Compte et al., 2018). The moderate omega (ω = 0.69) is consistent with what is expected for a heterogeneous multidimensional construct, in which abuse subtypes are conceptually related but not redundant.

d) Recognised limitation: We have added an explicit acknowledgement in the Limitations section that the moderate internal consistency of the scale and the impossibility of computing subtype-level reliability from the available data represent a methodological limitation, and we recommend that future studies collect item-level data to enable more granular psychometric analyses.

These additions are clearly marked in the revised manuscript.

Editor’s Comment 4. The manuscript used backward stepwise logistic regression, which may lead to unstable models or exclusion of theoretically important variables. Thus, author must briefly justify the modeling strategy and discuss its limitations in the limitations section.

Authors’ Response 4.

We appreciate this observation. We have added a brief methodological justification for the use of backward stepwise logistic regression in the “Statistical Analysis” subsection. The text now clarifies that this approach was chosen in an exploratory context, given the lack of a fully established theoretical model for selecting risk factors specific to older adult women in Southern Spain, and that the method allows the data to inform variable selection while retaining parsimony. We acknowledge that this approach may be sensitive to sample characteristics and could lead to the exclusion of theoretically relevant variables not supported by the specific sample.

In addition, we have expanded the Limitations section to explicitly discuss the limitations of backward stepwise regression, including the risk of model instability, potential overfitting in small-to-moderate samples, and the possibility that variables of theoretical importance may be excluded if their individual contribution does not reach the retention threshold. We recommend that future studies with larger samples consider block-entry or theory-driven regression strategies. These changes are marked in the revised manuscript.

Editor’s Comment 5. Ensure consistent use of the term “abuse against older adult women” throughout the manuscript.

Authors’ Response 5.

We thank the Editor for this comment. We have conducted a thorough review of the entire manuscript to standardise the use of the term “abuse against older adult women” where appropriate. Instances in which alternative formulations appeared (e.g., “elder abuse,” “old-age abuse,” or “mistreatment of older women”) have been revised for consistency, except in direct references to specific validated instruments (such as the “Geriatric Mistreatment Scale”) or in citations of prior literature where the original terminology is maintained. These changes are marked in the revised manuscript.

Editor’s Comment 6. Several socio-demographic variables (such as age, education, and income) were recoded into broad categories, which may reduce variability & statistical precision. The manuscript should briefly justify these recoding decisions and explain how the chosen categories align with previous research or contextual considerations.

Authors’ Response 6.

We agree that a justification for the recoding decisions was missing. We have added a paragraph in the “Socio-demographic information” subsection explaining that age, education, and income were grouped into broad categories to ensure sufficient cell sizes for statistical analysis (given a total sample of N = 209), to align with the categorisations used in comparable studies of older adult abuse in Spain and internationally, and to reflect contextually meaningful thresholds. For example, the binary age split at 80 years reflects the WHO classification of “old-old” adults and is widely used in gerontological research. The income categorisation is based on the IPREM (Indicador Público de Renta de Efectos Múltiples) threshold used in Spanish social policy. These additions are marked in the revised manuscript.

Editor’s Comment 7. Author may do minor language editing to remove repetitions.

Authors’ Response 7.

We have carefully reviewed the full manuscript to identify and eliminate repetitive passages. Redundant sentences and phrases—particularly in the Introduction and Discussion sections—have been revised or removed. The manuscript now reads more concisely, with no substantive content loss. All changes are marked in the revised version.

Reviewer #1:

I appreciate how the authors worked to improve the manuscript and address/integrate my comments/suggestions. Thank you for your sincere effort. I do notice that the internal reliability of the Geriatric Mistreatment Scale is not that great; but it is what it is based on the dataset you analyzed. If possible, please acknowledge this limitation in your manuscript and if you are able to justify briefly why you think your McDonald's omega value was only acceptable. Other than that, your revised version now has robust methods, clear presentation of results, and succint overview and discussion.

Authors’ Response 1.

We sincerely thank Reviewer 1 for their positive assessment of the revised manuscript and for their constructive observation regarding the internal reliability of the GMS. We have now explicitly acknowledged the moderate omega value (ω = 0.69) as a limitation in the Limitations section. In addition, we offer the following brief justification for why this value, while not optimal, is interpretively acceptable in the context of this study:

The GMS is a multidimensional scale covering five conceptually distinct types of abuse (physical, psychological, economic, sexual, and neglect). Unlike unidimensional scales, in which all items are expected to measure the same underlying construct and high internal consistency is a primary criterion, multidimensional instruments capturing heterogeneous constructs typically yield lower omega values. A moderate omega is expected—and is not necessarily a validity concern—when the scale is designed to screen across conceptually distinct subtypes rather than to measure a single latent trait. This is consistent with the original validation study of the GMS (Giraldo-Rodríguez & Rosas-Carrasco, 2013), and with the multidimensional structure of elder abuse more broadly.

Furthermore, the moderate omega in this sample may also reflect the relatively low prevalence of certain subtypes (e.g., neglect: 7.7%), which constrains the variance available for inter-item covariation and can systematically reduce omega estimates. These considerations have been added to the Limitations section of the revised manuscript.

Final Comment for Editors and Reviewers

We would like to express our sincere gratitude to the reviewers and the editors of PLOS ONE for their thoughtful feedback and constructive comments. We sincerely hope that the revisions undertaken adequately address the concerns raised and that the revised manuscript meets the standards of quality expected by the journal.

---

## [Editor Report · Decision Letter 2]

30 Apr 2026

Prevalence and risk factors of abuse against older adult women: a cross-sectional community study in Eastern Andalusia, Spain

PONE-D-24-58875R2

Dear Dr. Ricoy-Cano,

We’re pleased to inform you that your manuscript has been judged scientifically suitable for publication and will be formally accepted for publication once it meets all outstanding technical requirements.

Kind regards,

Vandana Dabla, Ph.D.

Academic Editor

PLOS One
---

## [Editor Report · Acceptance letter]

PONE-D-24-58875R2

PLOS One

Dear Dr. Ricoy-Cano,

I'm pleased to inform you that your manuscript has been deemed suitable for publication in PLOS One. Congratulations! Your manuscript is now being handed over to our production team.

Kind regards,

on behalf of

Dr. Vandana Dabla

Academic Editor

PLOS One